# ZnO Surface Doping to Enhance the Photocatalytic Activity of Lithium Titanate/TiO₂ for Methylene Blue Photodegradation under Visible Light Irradiation

**Anwar Iqbal [1,2,*], N. H. Ibrahim [1], Nur Ruzaina Abdul Rahman [3], K. A. Saharudin [3], Farook Adam [1], Srimala Sreekantan [3,*], Rahimi M. Yusop [2], N. F. Jaafar [1] and Lee D. Wilson [4,*]**

[1]   School of Chemical Sciences, Universiti Sains Malaysia, Minden 11800, Penang, Malaysia;
      nur_hanisah95@yahoo.com (N.H.I.); farook@usm.my (F.A.); nurfarhana@usm.my (N.F.J.)
[2]   School of Chemical Sciences and Food Technology, Faculty of Science and Technology,
      Universiti Kebangsaan Malaysia, Bangi 43600 UKM, Malaysia; rahimi@ukm.edu.my
[3]   School of Materials and Mineral Resources Engineering, Engineering Campus, Universiti Sains Malaysia,
      Nibong Tebal 14300, Penang, Malaysia; nur.ruzaina@yahoo.com (N.R.A.R.);
      khairularifah@gmail.com (K.A.S.)
[4]   Department of Chemistry, University of Saskatchewan, 110 Science Place, Saskatoon, SK S7N 5C9, Canada
[*]   Correspondence: anwariqbal@usm.my (A.I.); srimala@usm.my (S.S.); lee.wilson@usask.ca (L.D.W.)

**Abstract:** Wastewater contaminated with dyes produced by textile industries is a major problem due to inadequate treatment prior to release into the environment. In this paper, the ability of ZnO to enhance the interfacial photocatalytic activity of lithium titanate/TiO₂ (LTO/TiO₂) for the photodegradation of methylene blue (MB) under visible light irradiation (4.38 mW/cm²) was assessed. The ZnO-doped lithium titanate/TiO₂ (ZnO/LTO/TiO₂) was synthesized using a combination of hydrothermal and wetness impregnation methods. The high-resolution transmission electron microscope (HRTEM) and X-ray Diffraction (XRD) analyses indicate that the ZnO/LTO/TiO₂ contain several phases (ZnO, LTO, and TiO₂). The adsorption capacity of LTO/TiO₂ (70%) was determined to be higher compared to its photocatalytic activity (25%), which is attributed to the strong interaction between the Li and surface oxygen atoms with the MB dye molecules. The introduction of ZnO improved the photocatalytic ability of LTO/TiO₂ by 45% and extended the life span of ZnO/LTO/TiO₂. The ZnO/LTO/TiO₂ can be reused without a significant loss up to four cycles, whereas LTO/TiO₂ had reduced adsorption after the second cycle by 30%. The ZnO increased the surface defects and restrained the photo-induced electrons (e⁻) from recombining with the photo-induced holes (h⁺). Scavenging tests indicated that the hydroxyl radicals played a major role in the photodegradation of MB, which is followed by electrons and holes.

**Keywords:** titanate; visible light; methylene blue; lithium titanate; photocatalysis; ZnO

## 1. Introduction

Environmental pollution by textile industries continues to be prevalent despite the existence of regulatory requirements. Estimates indicate that ca. 60–70% of dyes remain on the fabric, whereas the residual dye may be released into water bodies as effluent without proper treatment [1]. Textile dyes can be highly carcinogenic and potentially mutagenic [2,3]. To address the health concerns, various water treatment methods such as flocculation, coagulation, precipitation, adsorption, filtration, and ion exchange methods have been utilized to remove the dyes from the wastewater. However, these methods are less attractive and impractical due to their technical complexity, high operating costs

(energy and materials), environmental fate, and high handling costs of the generated waste products. By contrast, advanced oxidation processes (AOPs) represent an alternative treatment strategy to address the disadvantages. AOPs were introduced in the 1980s for treating drinking water using $TiO_2$ as a photocatalyst. The use of $TiO_2$ in this respect offers advantages due to its high photostability, low cost, biological and chemical inertness, and non-toxic nature. AOPs typically utilize hydroxyl or sulfate radicals to remove the refractory organics, traceable organic contaminants, and certain inorganic pollutants to increase the wastewater biodegradability. At present, AOPs are widely used for treating wastewaters [4] where $TiO_2$ can be utilized effectively under UV light irradiation due to its wide band gap (3.2 eV). Upon irradiation by UV light, the conduction band excited the electrons from the valence band while leaving behind positively charged holes. The photo-induced electron-hole ($e^-/h^+$) pairs react with the water and oxygen to generate reactive oxygen species (ROS) such as superoxide anion radical ($\bullet O_2^-$), hydrogen peroxide ($H_2O_2$), singlet oxygen ($^1O_2$), and hydroxyl radical ($\bullet OH$). However, the rapid recombination of $e^-/h^+$ pairs hamper its practical utility as a photocatalyst. Even though sunlight is a cheap and a sustainable source of UV light, it is unreliable since the intensity of the sunlight depends on various geographic factors such as latitude, altitude, season, cloud thickness, and time.

Visible light constitutes about 43% of sunlight, and ca. 7% of the entire solar spectrum. Thus, it is beneficial to design a $TiO_2$-based photocatalyst that can efficiently photo-oxidize dyes in this spectral region. To achieve this goal and to decrease the recombination rate of $e^-/h^+$ pairs, $TiO_2$ has been doped with various metallic and non-metallic components. Various types of $TiO_2$-modified photocatalysts have been reported for the removal of various dyes [5–16]. Even though high levels of dye removal can be achieved, UV light or visible light coupled with specialized reactors is often required.

Lithium titanate (LTO) has been extensively applied as an anode in batteries [17], dye-sensitized solar cells [18], and super capacitors [19]. The small size of lithium cations makes it easier to incorporate into the lattice of $TiO_2$, either by replacing the O atoms or by occupying the interstitial sites. The doping method will extend the light-harvesting properties of $TiO_2$ to a higher wavelength; which in turn, enables the use of such catalysts under visible light irradiation for wastewater treatment.

Abdul Rahman et al. [20] reported the photocatalytic activity of LTO for the removal of methylene blue (MB) at 40 ppm under solar irradiation, where the prepared catalyst had high photocatalytic activity compared to Aeroxide® P25. However, it would be more attractive if the catalyst was functional under visible light conditions. In this contribution, we report on a photocatalyst containing the LTO and $TiO_2$ phase ($LTO/TiO_2$) synthesized via a hydrothermal method doped with ZnO ($ZnO/LTO/TiO_2$). This was achieved through a simple wetness impregnation method in order to increase the catalyst sensitivity toward visible light for the photodegradation of methylene blue (MB), as demonstrated by a simple homemade reactor equipped with 48-watt fluorescent lights with an intensity of 4.38 mW/cm². The photocatalytic activity of $ZnO/LTO/TiO_2$ improved significantly under visible light irradiation due to the abundant surface defects, which suppressed the recombination rate of the photoinduced $e^-/h^+$ pairs. In addition, the reduction in surface area and pore expansion was found to reduce the adsorption strength of the MB molecules on the catalyst surface. The stability of $ZnO/LTO/TiO_2$ was also found to be better than $LTO/TiO_2$. The advantages of this photodegradation system compared to those reported in the literature [5–16] are its cost-effectiveness, simplicity of the catalyst synthesis process, and the practical design features of this system. These features are envisioned to be appealing to the textile industry.

## 2. Materials and Methods

### 2.1. Materials

Chemicals used were hexadecyltrimethylammonium bromide (CTAB) (Acros Organics, 99%), lithium hydroxide (Acros Organics, 98%), titanium(IV) oxide, Aeroxide® P25 (Acros Organics, 95%), zinc oxide (R&M Chemicals, 99%), titanium(IV) oxide (Sigma-Aldrich, ≥99%), and methylene blue (MB)

(Sigma-Aldrich, ≥82%). The chemicals were of an analytical grade and used without any purification unless specified.

## 2.2. Synthesis of LTO/TiO₂

The synthesis of LTO/TiO₂ was carried out via a hydrothermal method, as described by Abdul Rahman et al. [20]. In a typical synthesis, 40 mL of 10 M LiOH, 4.00 g of titanium (IV) dioxide (TiO₂), and 0.800 g of hexadecyltrimethylammonium bromide (CTAB) was added to a 100-mL beaker with vigorous stirring using a magnetic stirrer for 30 min, which is followed by sonication for 1 h. The white suspension formed was aged under static conditions for 24 h. Then, the suspension was transferred into a 50-mL Teflon-lined stainless-steel autoclave and placed in an oven at 200 °C for 12 h. Upon cooling to 25 °C, the suspension was neutralized with 3 M $HNO_3$ and isolated under vacuum filtration along with washing with hot distilled water. The resulting LTO/TiO₂ powder was dried in an oven at 100 °C overnight, which was followed by calcination at 500 °C for 5 h.

## 2.3. Synthesis of ZnO/LTO/TiO₂

The synthesized LTO/TiO₂ was incorporated with ZnO via the wetness impregnation method by vigorously stirring the LTO/TiO₂ (0.97 g) and ZnO (0.03 g) in 50 mL of distilled water at 80 °C until the solvent evaporated. The photocatalyst was dried in the oven at 100 °C for 24 h, which is followed by calcination at 550 °C for 3 h.

## 2.4. Characterization

Fourier Transform Infrared (FT-IR) spectra were obtained using the KBr method and analyzed over the spectral range of 4000–400 cm$^{-1}$ (Perkin Elmer System 2000 FT-IR). The Brunauer-Emmett-Teller (BET) surface analysis was carried out using N₂-sorption porosimeter instrument (NOVA Quantachrome 2000e model) at 77 K with degassing at 150 °C for 10 h. UV-vis spectra were recorded over a spectral range of 200–800 nm with a Perkin Elmer Lambda 35 Spectrometer equipped with a diffuse reflectance attachment using KBr as the reference material. The X-ray Diffraction (XRD) analysis was carried out using a Bruker-D8 Advance diffractometer with Cu Kα radiation (0.1541 nm) over a range of 2θ-values (20° to 90°). The surface topology was analyzed using a Scanning Electron Microscope (SEM, Leica Cambridge S360). The X-Ray Photoelectron Spectroscopy (XPS) of the samples was carried out using an AXIS Ultra DCD model, Kratos, equipped with an Al Kα X-ray source-1486.6 eV at 10 mA and 15 kV to analyze a 300 μm × 700 μm area under $7.6 \times 10^{-9}$ Torr ultra-high vacuum environment in the sample chamber. The XPS is a surface sensitive and quantitative spectroscopic technique for measuring the oxidation state of the top 10 nm or top 20 layers of the catalysts' surface. To avoid the scattering of energy, the binding-energy scale was calibrated to C1s at 284.8 eV. The XPS spectrum were further deconvoluted using CasaXPS software. The morphology of the catalysts was analyzed using high resolution transmission electron microscope (HRTEM) 200 kV with field emission, TECNAI G2 20 S-TWIN FEI model. The photoluminescence (PL) method was used to determine the electron-hole recombination rate in the range of 200–900 nm using confocal Raman-PL IK3301R-G spectroscopy with He-Cd Laser Kimmon Koha (30.0 mW) as the excitation source.

## 2.5. Photocatalytic Degradation of Methylene Blue (MB) under Visible Light Irradiation

The photocatalytic degradation of MB was carried out in a homemade reactor equipped with two 24-watt fluorescent lamps fixed on the opposite sides of a box enclosure. The box was covered with aluminum foil to prevent excess light from entering the box. The light intensity in the box was measured to be 4.38 mW/cm$^2$. A schematic diagram of the reactor setup is shown in Figure S1 (cf. Supplementary Material). The photocatalyst (75 mg) was dispersed in 200 mL of a 10-ppm aqueous MB solution. The adsorption-desorption equilibrium was achieved under dark conditions for 2 h. The mixture was then exposed to the 48-watt fluorescent lamp system for 4 h with constant stirring. At specific time intervals, approximately 3 mL of the aliquot was collected through the transparent

pipe using a syringe and filtered with a 0.2-μm Sartorius Nylon Millipore filter. The aliquots were analyzed with a UV-vis spectrophotometer (Shimadzu UV-2600) at a fixed wavelength ($\lambda$ = 664 nm). The MB removal efficiency (%) was calculated using the formula below.

$$\text{Removal (\%)} = [(C_i − C_f)/C_i] \times 100\% \tag{1}$$

$C_i$ is the initial concentration (ppm) of MB and $C_f$ is the final concentration (ppm) of MB. The temperature inside the reactor was constant at 35 °C. For the reusability study, the spent photocatalyst was regenerated by washing with distilled water, which was followed by acetone washing, where the composite was subsequently dried at 100 °C prior to reuse.

## 3. Results

### 3.1. Characterization of LTO/TiO$_2$ and ZnO/LTO/TiO$_2$

3.1.1. X-Ray Diffraction and HRTEM Analyses

The X-ray diffraction (XRD) patterns of the photocatalysts are shown in Figure 1a. The diffraction lines at 25.4°, 37.7°, 38.1°, 38.9°, 48.4°, 54.2°, 55.4°, 63.0°, 69.2°, 70.6°, 75.4°, and 83.1° correspond to the (101), (103), (004), (112), (200), (105), (211), (204), (116), (220), (215), and (224) plane of the pure anatase phase having a tetragonal structure (ICSD: 98-003-7543). The diffraction lines observed at 31.8°, 34.4°, 36.2°, 47.5°, 56.6°, 62.9°, 66.4°, 68.0°, 69.0°, 72.6°, 77.0°, and 81.4° in the diffractogram of bare ZnO to indicate that the crystalline ZnO has the hexagonal wurtzite structure (ICDD file #98-002-9272) [21,22]. In the diffractogram of LTO/TiO$_2$, diffraction lines related to TiO$_2$ (anatase) can be observed together with the diffraction lines related to the LTO. LTO diffraction lines occur at 18.4°, 35.7°, and 43.4°, which correspond to the (111), (311), and (400) planes of a spinel cubic form of Li$_{0.33}$Ti$_{1.66}$O$_4$ (ICDD file 01-072-0426) [23]. Based on the XRD results, it is confirmed that the LTO and anatase coexist in LTO/TiO$_2$. In the diffractogram of ZnO/LTO/TiO$_2$, weak diffraction lines are assigned to the (100) and (110) plane of the ZnO detected at 31.7° and 56.6°, respectively. The low intensity diffraction lines and the absence of any other diffraction lines related to ZnO is likely due to the low concentration of ZnO. Based on the XRD analysis, the ZnO is proposed to exist as a separate phase together with LTO/TiO$_2$. The average crystallite size (D) of LTO/TiO$_2$ and ZnO/LTO/TiO$_2$ were estimated as 16.60 and 21.10 nm, respectively. The D value was calculated using Scherrer's formula.

$$D = K\lambda/\beta \cos\theta \tag{2}$$

K is a constant (0.94), $\lambda$ is the wavelength of X-ray radiation (CuK$\alpha$ = 0.1541 nm), $\beta$ is the line width at half maximum height (FWHM) of the diffraction line, and $\theta$ is the diffracting angle [24].

The morphology of LTO/TiO$_2$ and ZnO/LTO/TiO$_2$ observed from the HRTEM analysis and their corresponding interplanar distance (*d*-spacing) are presented in Figure 1b,c, respectively. The results in Figure 1b,c are in-line with the XRD analysis. The respective phases (ZnO, TiO$_2$, and LTO) that constitute the catalyst can be seen clearly in the micrographs. In Figure 1b, the *d*-spacing values are assigned to the (101) tetragonal anatase plane and (111) spinel cubic-shaped LTO/TiO$_2$ were 0.3516 nm and 0.2520 nm, respectively. The planes were shifted slightly in the presence of ZnO. The (101) hexagonal wurtzite-type ZnO plane was calculated to be 0.4838 nm in Figure 1c. Based on the XRD and HRTEM analysis, it is postulated that some of the TiO$_2$ may not have reacted with the LiOH during the hydrothermal synthesis process. As a result, a heterojunction was formed between TiO$_2$ and LTO. As for the ZnO/LTO/TiO$_2$, the various components (ZnO, LTO, and TiO$_2$) are annealed together to form a heterojunction during heat treatment. The estimated particle size of LTO/TiO$_2$ (88-143 nm) and ZnO/LTO/TiO$_2$ (65–88 nm) materials were calculated from the HRTEM images.

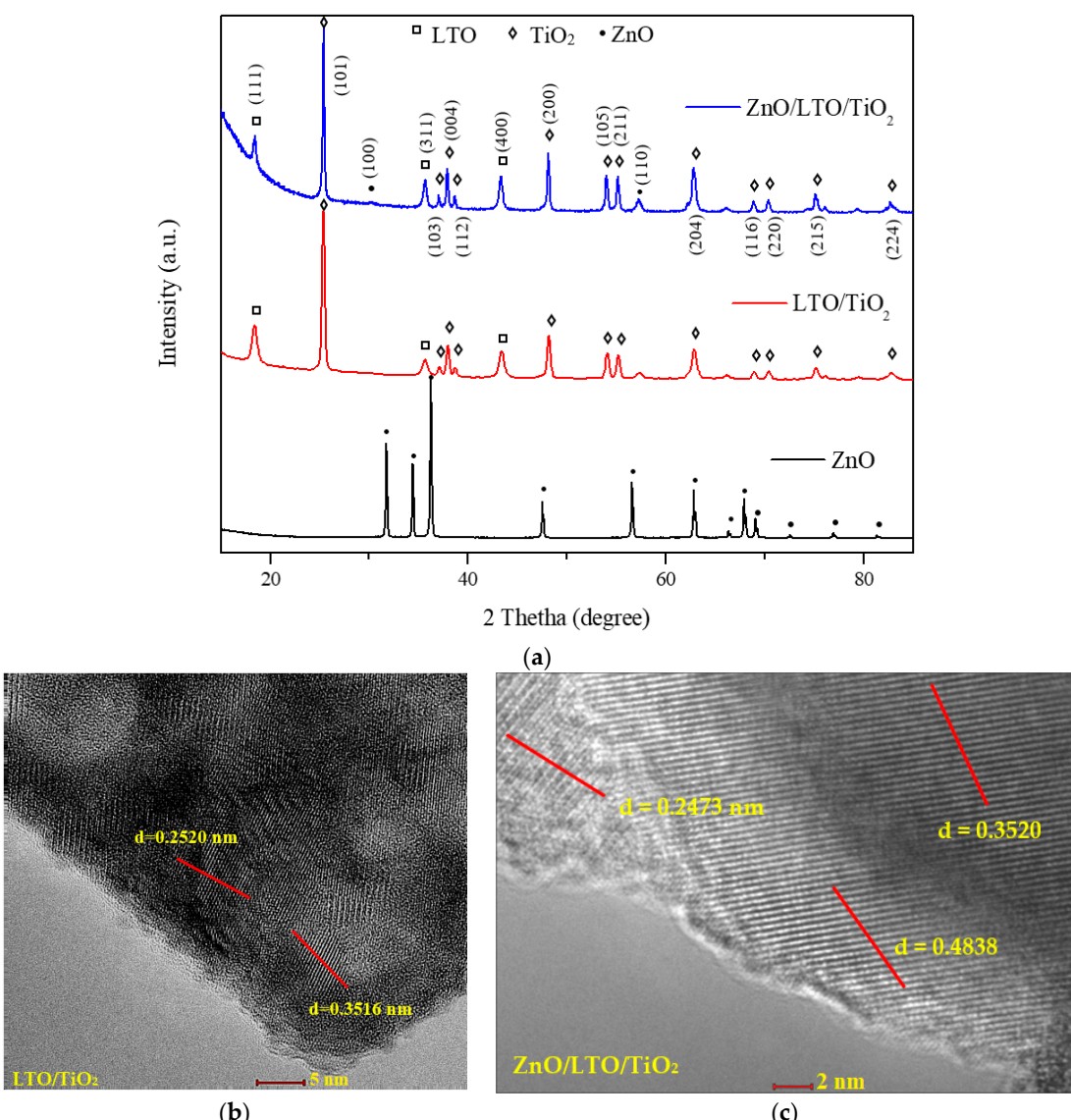

**Figure 1.** The XRD diffraction profiles (**a**) and high-resolution transmission electron microscope (HRTEM) images (with *d*-spacing) of lithium titanate (LTO)/TiO$_2$ (**b**) and ZnO/LTO/TiO$_2$ (**c**).

### 3.1.2. FTIR and N$_2$ Adsorption-Desorption Analyses

The FTIR spectra of the LTO/TiO$_2$ and ZnO/LTO/TiO$_2$ are shown in Figure 2a. The broad band in the spectral region around 450–840 cm$^{-1}$ is attributed to the symmetric stretching vibration of Ti-O-Ti and O-Ti-O flexion vibrations in the anatase phase [25]. This band appears broader in the ZnO/LTO/TiO$_2$ due to the presence of Zn-O stretching [26]. An envelope-shaped band ca. 3500 cm$^{-1}$ is a characteristic of O-H stretching vibration from adsorbed water [27]. The IR bands at 1441, 1500, and 1636 cm$^{-1}$ indicate the formation of Li$_2$CO$_3$, where it may have been formed upon the reaction of LiOH with atmospheric CO$_2$ in the presence of moisture [28].

The N$_2$ adsorption-desorption isotherms of the photocatalysts are shown in Figure 2b. The photocatalysts display a type IV isotherm with an H3 hysteresis loop, according to the International Union of Pure and Applied Chemistry (IUPAC) classification system. These features support the presence of mesopores, [29] where such composites are inferred to possess slit-shaped pores with non-rigid plate-like particles. The hysteresis loop of ZnO/LTO/TiO$_2$ appears to be narrower when compared to LTO/TiO$_2$ due to the deposition of ZnO within the mesopores.

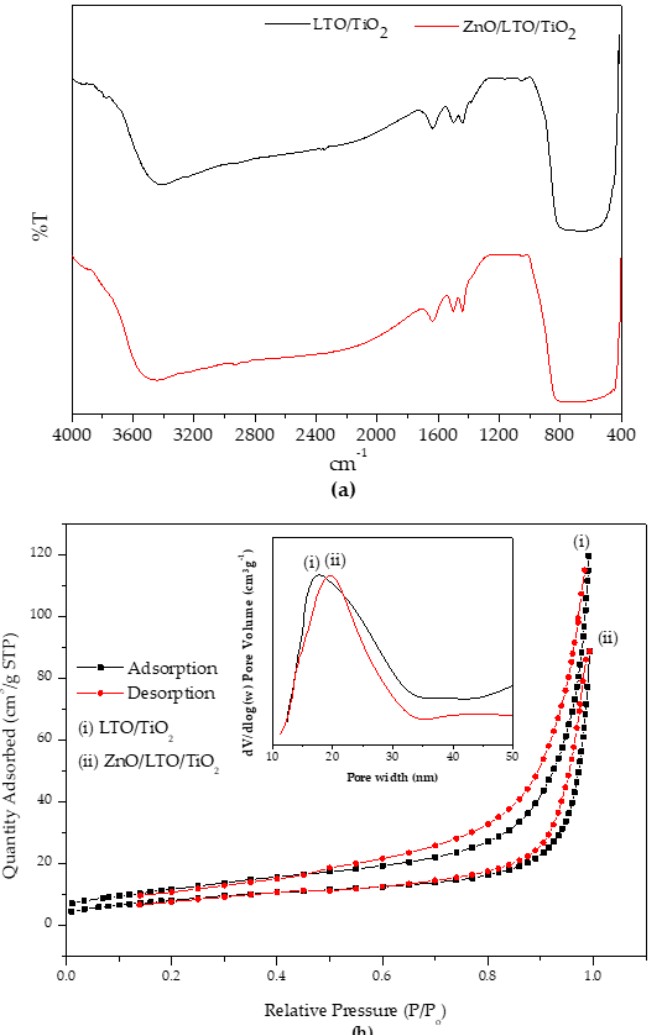

**Figure 2.** Characterization results for the LTO/TiO$_2$ and ZnO/LTO/TiO$_2$ composite materials: (**a**) IR spectra, and (**b**) N$_2$ adsorption-desorption profiles at 77 K. The inset shows the corresponding pore size distribution plot.

The pore volume of ZnO/LTO/TiO$_2$ (0.136 cm$^3$g$^{-1}$) was slightly lower as compared to LTO/TiO$_2$ (0.185 cm$^3$g$^{-1}$) due to the partial blockage of the mesopores and micropores by the ZnO nanoparticles. The pore blockage also reduced the overall porosity of the photocatalyst [30] including the surface area [31]. The Brunauer-Emmett-Teller (BET) surface area was reduced slightly from 38.8 m$^2$g$^{-1}$ (LTO/TiO$_2$) to 26.7 m$^2$g$^{-1}$ (ZnO/LTO/TiO$_2$) after the ZnO doping. The use of heat treatment to form heterojunctions between the ZnO and the LTO/TiO$_2$ may have ruptured the pore walls of the LTO/TiO$_2$, which leads to the formation of larger pores. The plot of Barrett–Joyner–Halenda (BJH) pore size distribution determined from the adsorption isotherm branch of the photocatalysts are shown as an inset in Figure 2b. The textural properties of the catalysts are also summarized in Table 1.

**Table 1.** Textural parameters from the N$_2$ adsorption-desorption analysis.

| Catalyst | BET Surface Area (m$^2$g$^{-1}$) | Average Pore Size (nm) | Average Pore Volume (cm$^3$g$^{-1}$) |
|---|---|---|---|
| LTO/TiO$_2$ | 38.8 | 17.2 | 0.1854 |
| ZnO/LTO/TiO$_2$ | 26.7 | 19.6 | 0.1363 |

### 3.1.3. Diffuse Reflectance-UV/Vis and Photoluminescent Analyses

The Diffuse Reflectance (DR)-UV/Vis spectra of LTO/TiO$_2$ and ZnO/LTO/TiO$_2$ are depicted in Figure 3a. Both photocatalysts exhibit intrinsic absorption in the ultraviolet region attributed to the band-band transition [32]. The band gap energy (E$_g$) of the composites was estimated from the intercept of a straight-line fitting to a plot of *[F(R)hv]$^{0.5}$* against *hv*, where *F(R$_\infty$)* is the Kubelka-Munk function and *hv* is the incident photon energy [33]. In Figure 3b, the calculated band gap energy for LTO/TiO$_2$ and ZnO/LTO/TiO$_2$ was estimated at 2.95 and 3.05 eV, respectively. Figure 3c compares the photoluminescence (PL) spectrum of LTO/TiO$_2$ and ZnO/LTO/TiO$_2$ photocatalysts. The spectra were collected under an excitation source of a 3-W cm$^{-2}$ He-Cd laser line (325 nm). Both photocatalysts have similarly shaped PL spectra due to the existence of the anatase phase (cf. Figure 1a). It is well known that the PL spectra of anatase are attributed to several types of phenomena: self-trapped excitons, oxygen vacancies, and surface states [34]. A PL band near 380 nm is attributed to the direct e$^-$/h$^+$ pair recombination, which corresponds to the band-gap energy of anatase (3.26 eV). The violet emission peak (450 nm, ~2.75 eV) arising from the indirect band edge allowed transitions and self-trapped excitons localized in TiO$_6$ octahedra. The self-trapped exciton is caused by the interaction of conduction band electrons localized on a Ti 3d orbital with holes in the O 2p orbital of TiO$_2$ [35]. The green emission at 520 nm and 560 nm are equivalent to ~2.38 and 2.21 eV, respectively. These emissions correspond to the deep-trap states far below the band edge emissions and are collectively called surface state emissions. In this process, surface defects such as non-bridging O$^{2-}$/O$^-$ trapped the photo-induced holes (h$^+$), which then migrated to a vacancy positioned deeper in the particle. The resulting deep hole trapped level is then located in the band gap of the photocatalyst, above the valence band. The photo-induced h$^+$ trapped in the deep oxygen vacancy level recombines with the photo-induced electrons (e$^-$) trapped in the shallow levels located below the conduction band, which gives rise to the visible luminescence [36]. The stronger PL intensity of ZnO/LTO/TiO$_2$ suggests that it contains more surface defects as compared to LTO/TiO$_2$.

### 3.2. Photocatalytic Degradation of Methylene Blue (MB)

The photocatalytic degradation of MB using LTO/TiO$_2$ and ZnO/LTO/TiO$_2$ was compared to P25. In Figure 4, when P25 was employed as a photocatalyst for MB removal, the adsorption level of MB was <10%, whereas 28% of the MB was removed when the system was irradiated with visible light. The isoelectric point (IEP) of P25 was reported to be in a range between pH 6.0 and 6.5 [37,38]. At this pH, the surface charge of P25 is neutral, whereas the surface of the P25 is negatively charged (TiO$^-$) above the IEP at alkaline conditions. By contrast, the surface of the P25 is positively charged (TiOH$_2^+$) at pH values below the IEP (acidic conditions) and the MB solution used for the screening was at pH 6. Due to the lack of electrostatic interaction between the cationic dye and the surface of the P25, less adsorption took place, which resulted in reduced photocatalytic activity.

The surface of the LTO/TiO$_2$ was determined to be positively charged since its IEP was pH 8.06. Supposedly, the positively-charged surface should repel the cationic dye (MB). However, the adsorption of MB on the surface of LTO/TiO$_2$ was calculated to be 70%. This observation is attributed to the presence of Li atoms and surface oxygen atoms, where MB can coordinate with the Li atoms. Moreover, the surface of LTO/TiO$_2$ has an abundance of oxygen atom surface sites that can undergo electrostatic interactions with the positive R$_2$ = N$^+$ = R groups of MB dye molecules [39]. These factors will increase the adsorption of MB with the surface sites of LTO/TiO$_2$. Due to the strong adsorption of MB, ca. 25% of the total removal (95%) is attributed to the photocatalytic activity. When the surface of LTO/TiO$_2$ was saturated with the MB dye, the penetration of visible light to excite the electrons will be hampered and lowers the photocatalytic activity. Thus, it is concluded that the removal of MB using LTO/TiO$_2$ was predominantly due to adsorption rather than photocatalysis.

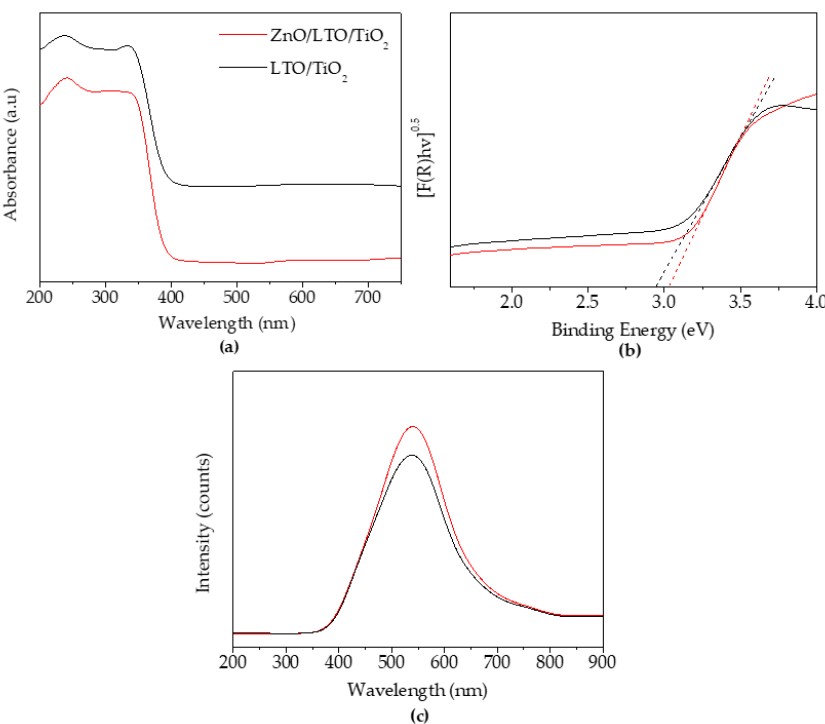

**Figure 3.** (**a**) The UV-vis reflectance spectra of the composites, (**b**) the corresponding plots of $[F(R\infty)h\nu]^{0.5}$ versus binding energy, and (**c**) the photoluminescence (PL) spectra of the composites.

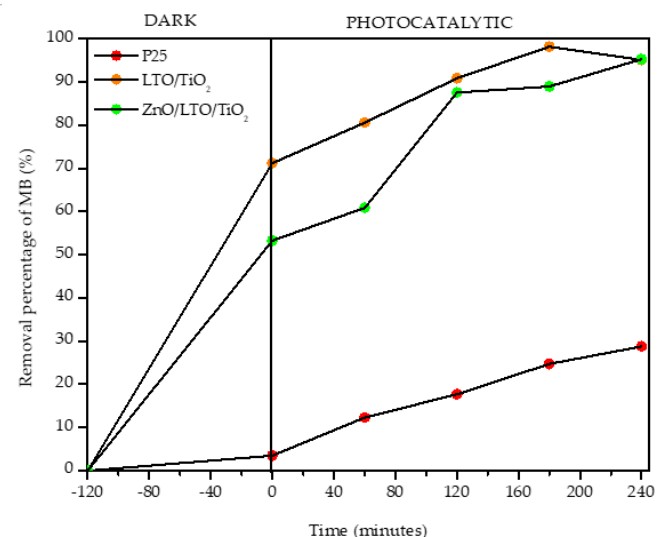

**Figure 4.** The photocatalytic removal of MB using P25, LTO/TiO$_2$, and ZnO/LTO/TiO$_2$. (C$_i$ = 10 ppm, pH 6, mass of composite = 75 mg, t = 4 h).

The photocatalytic removal of MB increased by 45% in the presence of ZnO/LTO/TiO$_2$. By comparison, the total removal of MB at the end of reaction time was 95%. The doping of ZnO enhanced the photocatalytic activity of LTO/TiO$_2$ in several ways. The surface area of ZnO/LTO/TiO$_2$ was measured to be lower than LTO, whereas the pore size increased (cf. Table 1). These characteristics reduced the adsorption strength of the MB onto the catalyst surface. Hence, the probability for the reactive oxygen species (ROS) to react with the MB molecules increased since the catalyst was able to absorb light more effectively due to reduced adsorption of MB on the catalyst surface. In addition, the excess surface defects created by ZnO will suppress the recombination rate of photo-induced e$^-$/h$^+$ pairs, which results in greater ROS generation. Furthermore, the particle size of ZnO/LTO/TiO$_2$

(65–88 nm) is smaller when compared to LTO/TiO$_2$ (88–143 nm), which favour the formation of ROS since the photo-induced e$^-$/h$^+$ pairs migrate faster to the active surface sites of catalysts with a smaller particle size [40].

The pH of the MB solution is an important factor in determining photocatalytic activity since the pH alters the mode of interaction between the organic dye pollutant with the catalyst surface, along with the radical charges formed during the reaction [41]. The photocatalytic degradation of MB was carried out from pH 2 to 12 and the MB removal plots are shown in Figure 5a. In acidic media, the surface of the ZnO/LTO/TiO$_2$ is postulated to be extremely positive due to the presence of excess H$^+$ ions. Under these conditions, the positively-charged surface will repel the positively-charged MB dye molecules. Due to the weak electrostatic interaction, only 17% of MB dye was removed. Since the environment becomes more alkaline, the surface progressively becomes negatively charged [42] and enhances electrostatic interactions between MB and the photocatalyst surface, which results in greater adsorption capacity. The MB removal at pH 4 was 82%, whereas greater removal (95–99%) occurred at a pH of 6–12. Due to an investigation of different dosages of the composite, the pH of the MB solution was fixed at a pH of 6.

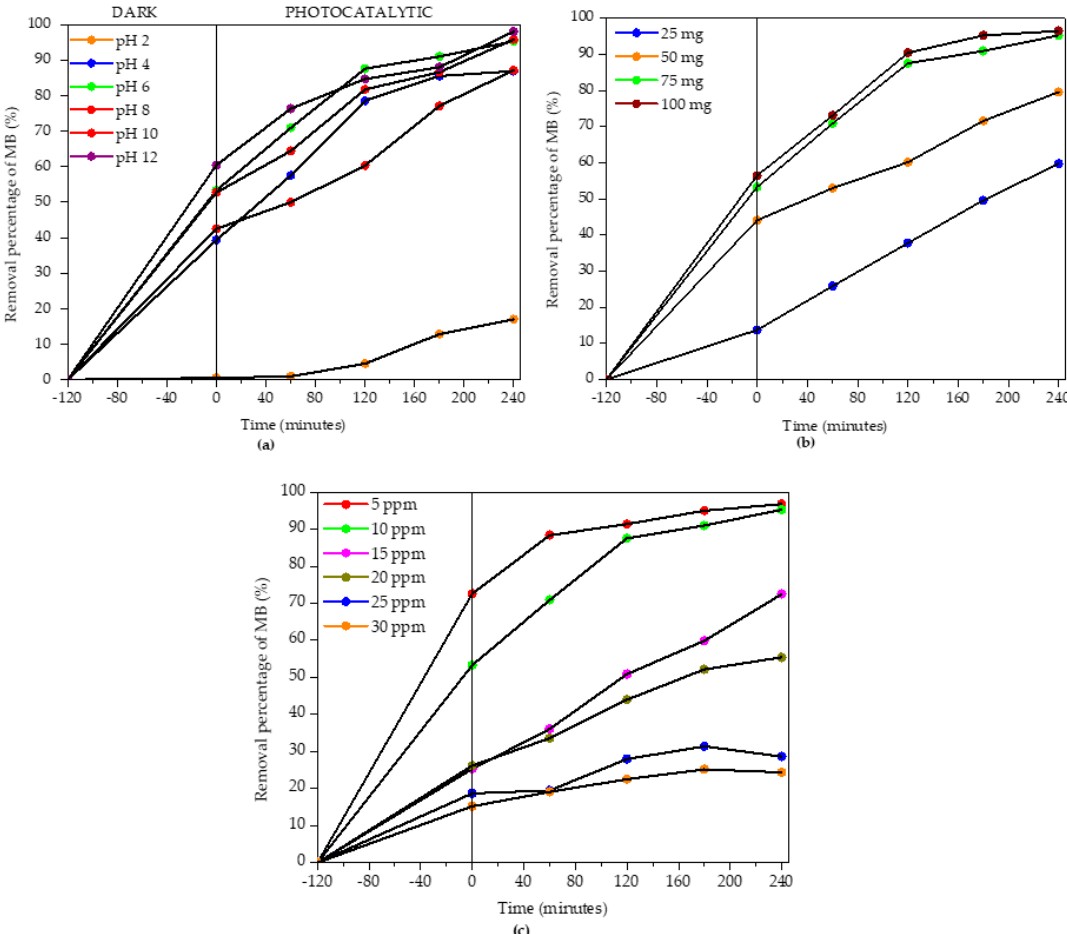

**Figure 5.** The effect of (**a**) pH (C$_i$ = 10 ppm, mass of photocatalyst = 75 mg, t = 4 h), (**b**) catalyst dosage (C$_i$ = 10 ppm, pH of 6, t = 4 h), and (**c**) photodegradation kinetics study using ZnO/LTO/TiO$_2$ at different MB concentration (pH 6, t = 4 h, 75 mg ZnO/LTO/TiO$_2$).

An investigation of the catalyst dosage is essential for optimizing the effective removal of MB. Figure 5b shows the influence of ZnO/LTO/TiO$_2$ dosage on the photodegradation of MB. It is observed that the photodegradation of MB increased as the photocatalyst dosage was increased from 0.025 to 0.075 g L$^{-1}$. The greater number of active sites enhanced the generation of ROS and provided a greater

surface area for dye adsorption [43–45]. A further increase of the catalyst dosage to 0.1 g L$^{-1}$ did not enhance the photodegradation process, whereas the turbidity of the solution for the reaction increased at this catalyst dosage due to reduced light penetration [45]. To offset such inhibitory effects on the photodegradation process, an optimum catalyst dosage was determined to be 0.075 g L$^{-1}$.

The photodegradation kinetics was studied using ZnO/LTO/TiO$_2$ at various MB concentrations at optimized conditions, where the photocatalyst was able to efficiently remove up to 10 ppm MB (95.3%). By contrast, the efficiency of the composite was reduced when a higher dye concentration was used since the multilayers of adsorbed dye attenuate the light penetration onto the photocatalyst surface. Thus, extensive dye adsorption is likely to inhibit the formation of holes, electrons, and ROS. To analyze the photocatalytic kinetic parameters for the dye degradation in the presence of the composite, a first-order kinetic model was used, as follows.

$$\ln (C_o/C_t) = kt \tag{3}$$

k is the pseudo first-order rate constant, C$_o$ and C$_t$ are the initial and final dye concentration (ppm), and t is the time (h) [46]. Figure 5c shows the pseudo-first-order rate constant using the Langmuir-Hinshelwood model of different MB concentration. It was found that 20 ppm, 25 ppm, and 30 ppm are nonlinear profiles that have exceptionally low values of the correlation coefficient (R$^2$ ≤ 0.9), as listed in Table 2. Thus, the first-order kinetic model does not fit well for the whole range of contact times.

**Table 2.** The pseudo-first-order rate constant using the Langmuir-Hinshelwood model at variable methylene blue (MB) dye concentration.

| Initial Concentration, (ppm) | Rate Constant, k (h$^{-1}$) | Correlation Coefficient (R$^2$) | MB Decolorization (%) |
|---|---|---|---|
| 5 | 0.585 | 0.9395 | 96.8 |
| 10 | 0.586 | 0.9412 | 95.3 |
| 15 | 0.246 | 0.9788 | 72.5 |
| 20 | 0.0931 | 0.7407 | 45.4 |
| 25 | 0.0607 | 0.7911 | 28.6 |
| 30 | 0.006 | 0.0739 | 24.4 |

Meanwhile, 5 ppm, 10 ppm, and 15 ppm showed improved linearity, where R$^2$ ≥ 0.90, and indicates that this model fits well over the range of contact times. Thus, it shows a significant and favorable effect of ZnO/LTO/TiO$_2$ on the photodegradation of MB. The rate constant increases as the MB concentration decreases, where no significant changes are observed at 5 ppm. This might occur due to the higher MB adsorption at lower concentrations. In conclusion, ZnO/LTO/TiO$_2$ at a fixed dosage (0.075 g L$^{-1}$) led to photocatalytic degradation (95%) of MB solution (10 ppm) at a pH of 6 under visible light irradiation within 4 h.

### 3.3. Free Radical Scavenging Study

Scavengers were added to identify the reactive species that contribute to the photocatalytic degradation of MB [47]. The scavengers used to scavenge holes, electrons, and hydroxyl radicals include disodium oxalate, silver nitrate, and acetonitrile, respectively. The MB removal plot in the presence of the scavengers is shown in Figure 6a. The MB removal was reduced to 83.8% in the presence of disodium oxalate, which suggests that holes have a minor contribution to the degradation process. The MB removal was suppressed to 58.1% when silver nitrate was used as an electron scavenger, whereas ca. 90% of the photodegradation was suppressed when acetonitrile was added to scavenge the HO• radicals. Based on the suppressed level of photodegradation, it is concluded that the main species involved in the MB photocatalytic degradation was hydroxyl radicals, which was followed by electrons and holes.

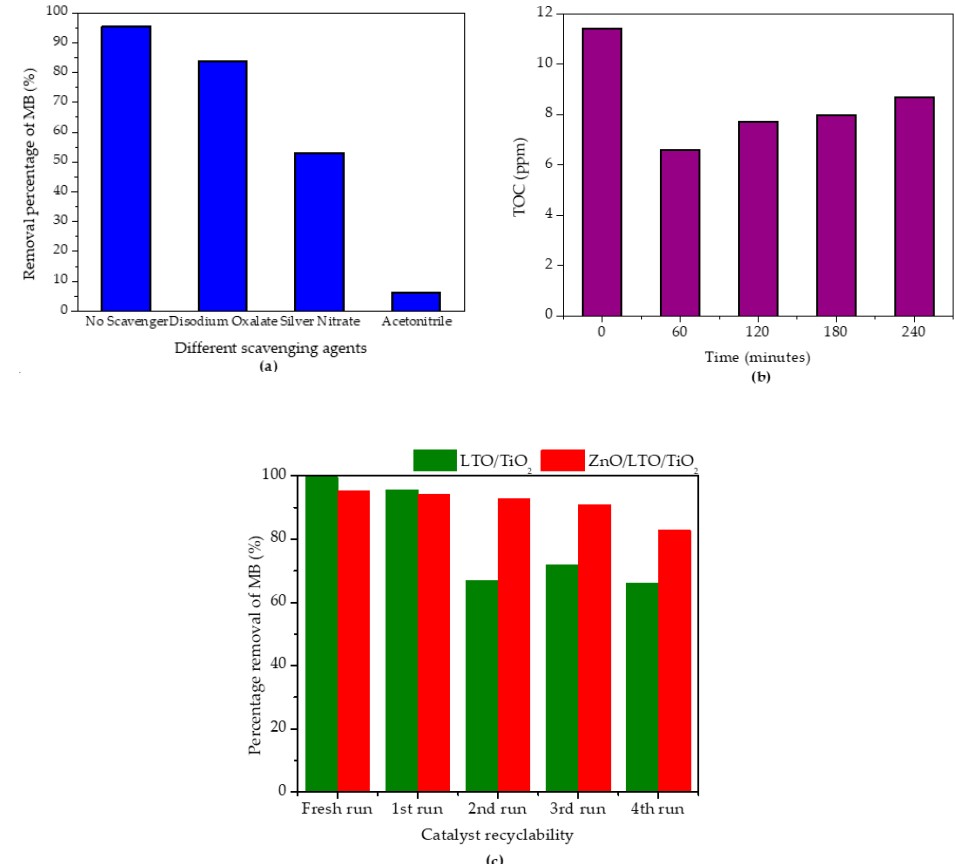

**Figure 6.** (**a**) Effect of different scavengers on the photocatalytic degradation of MB, (**b**) the total organic carbon (TOC) reduction level under visible light, and (**c**) stages of reusability of ZnO/LTO/TiO$_2$ and LTO/TiO$_2$ ($C_i$ = 10 ppm, pH 6, t = 4 h, 75-mg catalyst).

### 3.4. Mineralization Study

Complete decolorization of the MB does not indicate the complete mineralization of the dye to CO$_2$ and H$_2$O. The value of the total organic carbon (TOC) is linked to the total concentration of organic factors in the solution and the decrease of TOC value reflects the degree of mineralization [48]. The TOC for the photocatalyst under visible light irradiation is shown in Figure 6b. After irradiation for 4 h, only 41% of mineralization was achieved, which indicates that the structure of MB was degraded into smaller molecular fragments.

### 3.5. Reusability Study

Catalyst stability or reusability is an important factor in heterogeneous catalysis. The stability of ZnO/LTO/TiO$_2$ was tested by reusing the photocatalyst for up to five cycles (one fresh use and four reuse cycles) in Figure 6. The MB removal plot during the reusability study is given in Figure 6c along with a comparison of the reusability of LTO/TiO$_2$ is also shown. The ZnO/LTO/TiO$_2$ was stable when reused up to four times and dropped to 82.8% during the fifth cycle, whereas the removal of MB dropped ~30% when LTO/TiO$_2$ was reused for the second time, but it remained constant up to the fifth cycle. As the LTO/TiO$_2$ was reused for the second cycle, more dye molecules were adsorbed strongly onto the catalyst surface, which reduced visible light penetration, and, thereby, inhibited electron excitation. During the third cycle and beyond, MB removal occurred mainly by adsorption. The doping of ZnO prevented the dye molecules from being strongly adsorbed onto the surface of LTO/TiO$_2$. The weak interaction caused the degradation products to desorb rapidly after the reaction and permit continuous absorption of visible light. As a result, the ZnO/LTO/TiO$_2$ can be reused multiple times as

compared to LTO/TiO$_2$. The FTIR analysis of the spent ZnO/LTO/TiO$_2$ (Figure S3) did not indicate the deterioration of the catalyst functional groups. The FTIR analysis also indicates that the adsorbed MB or its photodegraded products were removed by washing with water and acetone.

The practicality of ZnO/LTO/TiO$_2$ to be applied in a real situation was compared to several photocatalysts reported in the literature. From Table 3, the superiority of ZnO/LTO/TiO$_2$ compared to the reported catalysts is its ability to function under low intensity visible light (4.38 mW/cm$^2$). Since only a readily available 48 W fluorescent lamp is required, the overall cost of this system is affordable for use, especially for a small-scale textile manufacturer.

**Table 3.** Comparison with recently reported ZnO and TiO$_2$ modified photocatalysts for degradation of methylene blue (MB) under visible light irradiation.

| Catalyst | Dye Removal (%) | Dye (ppm) | Time (min) | Reusability (Cycles) | Irradiation Source | Reference |
|---|---|---|---|---|---|---|
| TiO$_2$-20% graphene | 98.8 | 10 | 100 | Three | Halogen-tungsten lamp (500 W) | [49] |
| Ag/TiO$_2$/rGO | 79 | 10 | 240 | NA | Fluorescence xenon lamp (200 W) | [50] |
| Cu$_2$O/TiO$_2$ | 100 | $5.0 \times 10^{-5}$ M | 90 | Three | Xenon light (300 W) | [51] |
| ZnO/3% SrO | 100 | 10 | 6 | Four | Halogen lamp (500 W) | [52] |
| W-TiO$_2$/RGO | 99.8 | 10 | 90 | Four | Xenon lamp (400 W) | [53] |
| ZnO/1% CuO | 95.52 | 10 | 5 | Four | Halogen lamp (500 W) | [54] |
| CuPc/TiO$_2$ | 100 | 20 mM | 150 | Five | Xenon lamp (150 W) | [55] |
| 0.5%NiO/m-TiO$_2$ | 90 | 20 | 150 | Five | Visible light GE lamp (400 W) | [56] |
| Fe-Cd (2%):ZnO | 82 | 20 | 140 | Five | Xenon lamp (300 W) | [57] |
| $_{0.4}$ZnO:$_{0.6}$TiO$_2$ | 90 | 10 | 75 | NA | Tungsten lamp (100 W) | [58] |
| ZnO/LTO/TiO$_2$ | 95 | 10 | 240 | Four | Fluorescent lamps (48 W) | This work |

Note: NA—not applicable.

### 3.5.1. XPS Analysis of ZnO/LTO/TiO$_2$

The surface element composition and the chemical state for ZnO/LTO/TiO$_2$ was determined using XPS analysis. The Zn-2p, O-1s, Ti-2p, Li-1s, and C-1s XPS spectra are given in Figure S2a–e, respectively (cf. Supplementary Material). The deconvolution of Zn 2p core level (Figure S2a) shows two energy bands, centered at 1023.6 eV and 1021.6 eV, which correspond to the spin orbit of Zn 2p$_{3/2}$ and Zn 2p$_{1/2}$, respectively. The appearance of these bands is due to Zn$^{2+}$ species of ZnO [59,60]. Wu et al. [61] reported that the substitution of Ti$^{4+}$ ion by Zn$^{2+}$ ion during sample preparation presented a challenge since the radius of Zn$^{2+}$ ion is larger than Ti$^{4+}$ ion. This observation further confirms that ZnO exists as a separate phase. Figure S2b shows the deconvoluted O 1s core level XPS spectrum of ZnO/LTO/TiO$_2$. The band at 531.5 eV is associated with the oxygen atom of the carbonate species, oxygen atoms in the oxygen-deficient (V$_o$) regions of ZnO, hydroxyl groups of physiosorbed water on TiO$_2$, and oxygen atoms of the Li-O bond [62–64]. Another band ~529.7 eV corresponds to the oxygen atom bound to metals such as Li, Zn, and Ti [65,66]. The deconvolution of Ti 2p core level (Figure S2c) resulted in two bands at 458.8 eV and 464.6 eV. These bands correspond to the spin orbit of Ti 2p$_{3/2}$ and Ti 2p$_{1/2}$ of TiO$_2$ [67]. Figure S2d shows the deconvoluted core level XPS spectrum of Li 1s. The band at 55.1 eV relates to Li$_2$CO$_3$, whereas the band at 54.2 eV is due to the Li-O bond of LTO [68]. The deconvolution of the C1s core level reveals three signatures (Figure S2e). The band at ~284.6 eV is referenced to the adventitious carbon, which is typical of carbon contamination from the environment [69]. Another band ~285.8 eV corresponds to the carbon atom bound only to C or H atoms

that originate from hydrocarbon contaminants from atmospheric carbon and organic residues from the chemical synthesis [68]. The third band at ~288.7 eV corresponds to the formation of C = O [70].

### 3.5.2. Proposed Reaction Mechanism

A potential photodegradation mechanism was proposed that is based on the physicochemical data, MB photodegradation, scavenging tests, and the band structures. The relationship between the conduction band (CB) and valence band (VB) potential of the catalyst was determined using the following equation.

$$E_{VB} = X - E_C + 0.5E_g \tag{4}$$

where $X$ is the absolute electronegativity of the semiconductor ($X$ is 5.81 eV for $TiO_2$, 5.95 eV for ZnO, and 1.94 eV for LTO). The $E_c$ is the energy of free electrons based on a hydrogen scale (4.5 eV), whereas $E_g$ is the band gap energy of the semiconductor. The CB potential was determined through the following equation.

$$E_{CB} = E_{VB} - E_g \tag{5}$$

Since the LTO has a narrow band gap (2.95 eV), the photoinduced $e^-$ in its VB will be excited to the CB when irradiated with the visible light. As the $e^-$ was excited, the same number of holes ($h^+$) will be created in its VB. The $e^-$ will then be transferred from the CB of LTO to the CB of $TiO_2$ and, then, to the CB of ZnO. The CB of ZnO and $TiO_2$ can be described as a pool to contain the photoinduced $e^-$ that hold them from recombining with the $h^+$ domains. In addition, the $e^-$ can react with dissolved $O_2$ in aqueous solution to form $O_2 \bullet^-$. This strong oxidative species combines with the $H^+$ from the solution to form $H_2O_2$. The $e^-$ can further react with the $H_2O_2$ to form $\bullet OH$. The hydroxyl radical plays a key role in MB degradation. Besides, the $h^+$ is also capable of generating $\cdot OH$ from $HO^-$ since the potential of the CB of ZnO and $TiO_2$ are more positive than $\bullet OH/OH^-$ (2.40 eV vs. NHE). The proposed mechanism is in-line with the scavenging test results, which indicate that the $\bullet OH$ was mainly involved in the photodegradation of MB dye. This is followed by $e^-$ and $h^+$. The photodegradation of MB using ZnO/LTO/$TiO_2$ is illustrated in Figure 7.

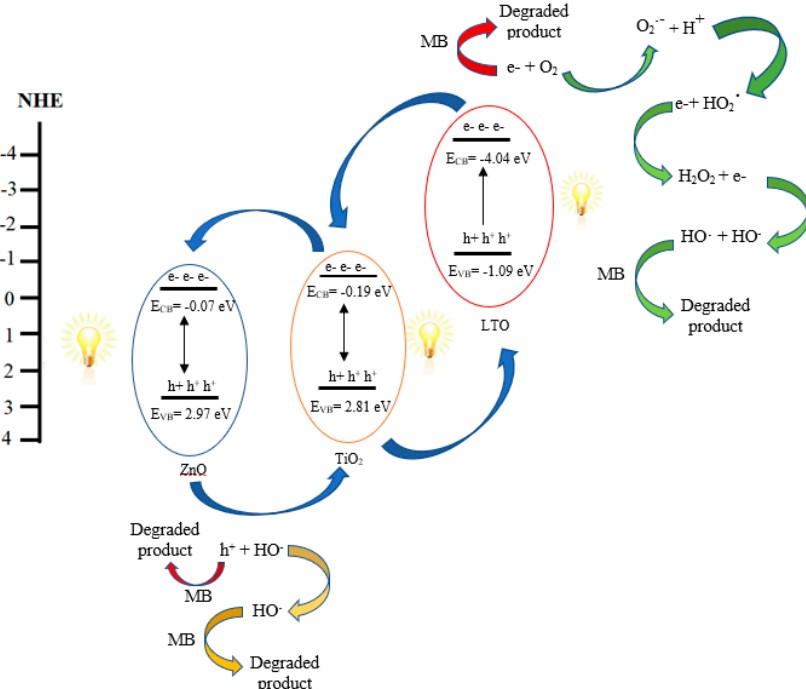

**Figure 7.** Schematic illustration of the charge transfer pathway during photodegradation of MB over ZnO/LTO/$TiO_2$ under visible light irradiation.

## 4. Conclusions

This study assessed the feasibility of ZnO in enhancing the sensitivity of a lithium titanate/TiO$_2$ (LTO/TiO$_2$) catalyst using visible light. The doping of ZnO weakens the adsorption between MB and the surface of the catalyst (ZnO/LTO/TiO$_2$). In addition, the surface defects created in the presence of ZnO suppress the recombination of e$^-$/h$^+$ pairs by acting as a pool to contain the photoinduced e$^-$. The highest photocatalytic degradation of MB (95%) was achieved when 0.075 g L$^{-1}$ ZnO/LTO/TiO$_2$ was used at a pH of 6.0. The reaction was carried out for 2 h in the dark and 4 h under visible light irradiation (4.38 mW/cm$^2$). The doping of ZnO also increased the reusability of the composite since MB and its photodegradation products were not strongly adsorbed onto the catalyst surface. The hydroxyl radicals were identified as the driving force in the photocatalytic process, which were followed by electrons and holes. The results of this research are likely to contribute to an improved design of advanced catalyst systems for the photodegradation of waterborne dyes to address the growing need for wastewater treatment of industrial effluent.

**Supplementary Materials:** The following are available online at http://www.mdpi.com/2571-9637/3/3/22/s1. Figure S1: Schematic diagram of the photocatalytic reactor used in the photodegradation reaction of MB. Figure S2: The XPS spectra of (a) Zn 2p, (b) O 1s, (c) Ti 2p, (d) Li 1s, and (e) C 1s of ZnO/LTO/TiO$_2$, and Figure S3: FTIR spectra of spent ZnO/LTO/TiO$_2$.

**Author Contributions:** Conceptualization, A.I., F.A., and S.S. Methodology, A.I., F.A., and S.S. Validation, A.I., K.A.S., and S.S. Formal analysis, A.I., N.H.I., N.R.A.R., and K.A.S. Investigation, A.I., N.H.I., N.R.A.R., and K.A.S. Resources, A.I., F.A., S.S., R.M.Y., N.F.J., and L.D.W. Data curation, A.I., N.H.I., N.R.A.R., and K.A.S. Writing—original draft preparation, A.I., N.H., N.R.A.R., S.S., K.A.S., R.M.Y., and L.D.W. Writing—review and editing, A.I., N.H.I, N.R.A.R., S.S., K.A.S., R.M.Y., F.A., and L.D.W. Visualization, A.I., S.S., and F.A. Supervision, A.I., S.S., and F.A. Project administration, A.I. Funding acquisition, A.I., F.A., S.S., R.M.Y., and L.D.W. All authors have read and agreed to the published version of the manuscript.

**Funding:** The Ministry of Education Malaysia (Higher Education) and Universiti Sains Malaysia Research University Grant (RUI) (1001/PKIMIA/8011083), FRGS Grant (203/PKIMIA/6711790), and TRGS Grant (203/PKIMIA/679001) funded this research. The Geran Universiti Penyelidikan Universiti Kebangsaan Malaysia (GUP-2019-045) and Universiti Sains Malaysia Short Term Grant (304/PKIMIA/6313215) also partially funded this research.

**Acknowledgments:** The authors would also like to thank the Ministry of Education Malaysia (Higher Education) and Universiti Sains Malaysia for the Post-Doctoral Training Fellowship awarded to Anwar Iqbal.

**Conflicts of Interest:** The authors declare no conflict of interest.

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
