# Peer review of "ZnO Surface Doping to Enhance the Photocatalytic Activity of Lithium Titanate/TiO2 for Methylene Blue Photodegradation under Visible Light Irradiation"

_surfaces, doi:10.3390/surfaces3030022_

Round 1
Reviewer 1 Report
This manuscript reported the construction of the ZnO/LTO/TiO2 heterostructures for photocatalytic pharmaceuticals degradation. The structure and catalytic performance of the catalysts were evaluated by some suitable characterizations. The introduction of ZnO nanoparticles improved the photocatalytic ability of LTO/TiO2. Therefore, the manuscript is worthy of publication after a minor revision. The following points should be addressed in revised version.
- For the identification the main reactive species, the authors conducted the trapping experiments. Photoinduced generation of reactive radicals of catalysts should be further detected via electron spin resonance (ESR) measurement and compared with LTO/TiO2.
- The spatial separation of the reduction and oxidation cocatalysts could reduce the recombination of the photogenerated charges. The photoelectrochemical measurements, such as i-t curves and EIS analysis, can evaluate the recombination of the photoinduced charges. But the photoelectrochemical measurement are lack in this manuscript, the authors should further investigate the photoelectrochemical measurements.
3. For the photocatalytic degradation, the stability and durability are important evaluation indicator of the photocatalyst. The authors conducted the cycling experiments, but the catalyst structure after the long-term circulation measurement should be further investigated.
Reviewer 2 Report
- Authors should give more clear details about purpose in the introduction section. The novelty or significance of this work is not clearly stated. What are the advantages of the method over previous strategies?
- Typographical errors must be corrected throughout the manuscript.
- All the figures and its corresponding captions must be defined clearly for better understanding for the readers. Some figures are expanded. Also, the author must pay attention to legend of the figures. It must be revised properly. For instance, Figure 2b is not defined in captions.
- In Figs 4-6, time x-axis unit must be minutes.
- In Figure 2b, pore size distribution must be provided.
- Page 8, line 306-310, lines are highlighted.
- Line 217-219- the sentence need to be revised.
- Tables and equation in the manuscript must be presented with respect to journal format.
- Regarding the low visible light irradiation, authors have mentioned 48 watts in abstract and in page 3 authors have mentioned 43.8 W/m-2. Authors have to describe the light intensity in mW/cm2.
- Authors have to clearly differentiate between LTO/TiO2 and LTO composite in the manuscript.
- Electrochemical impedance spectra measurements must be carried out to prove the conductivity of the materials.
- XPS spectra of bare ZnO, LTO/TiO2 spectra must be compared. More detailed spectra of composite must be provided with appropriate legend and discussed clearly.
- During the results presentation, please provide a comparison with previous similar materials. Why this catalyst should be better than the previous catalysts?
- Figure 4 x-axis unit must be revised.
- The English of the manuscript must be improved before resubmission. The author should pay particular attention to English grammar, spelling, and sentence structure so that the expression of the study is clear to reader.
- Explain better the mechanism of MB degradation. Why by incorporating the ZnO to LTO/TiO2, the degradation was improved? What about the mechanism of degradation when ZnO and LTO were used alone?
- More related papers should be cited/referrred 3390/coatings10050500; 10.1016/j.jphotochem.2019.02.020; 10.1016/j.jallcom.2020.154281; 10.1016/j.colsurfa.2019.124294; 10.1088/1361-6528/ab268a/meta; 10.1016/j.optmat.2018.05.022
Author Response
Author Response to Reviewer comments on MS ID: surfaces-845611
We would like to thank the reviewers for the thoughtful comments and constructive suggestions to improve the quality of this manuscript. Any changes that were made to the manuscript is highlighted in yellow. Following are our response to the reviewers concerns point by points.
Reviewer #2
Comments and Suggestions for Authors
- Authors should give more clear details about purpose in the introduction section. The novelty or significance of this work is not clearly stated. What are the advantages of the method over previous strategies?
Response:
Thank you for the comment and the suggestion. The significance of this work has been included in Section 1, page 2, last paragraph. The advantages of this method over previous strategies are discussed in page 13, last paragraph. A few example of catalysts reported in the literature are shown in Table 3. The changes are highlighted in yellow.
- Typographical errors must be corrected throughout the manuscript.
Response: Thank you for the comment.The manuscript has been comprehensively edited for language as recommended.
- All the figures and its corresponding captions must be defined clearly for better understanding for the readers. Some figures are expanded. Also, the author must pay attention to legend of the figures. It must be revised properly. For instance, Figure 2b is not defined in captions.
Response: Thank you for the comment. All the figures and its corresponding captions have been revised.
- In Figs 4-6, time x-axis unit must be minutes.
Response: Thank you for the comment. The x-axis of Figs 4-6 has been changed to minutes.
- In Figure 2b, pore size distribution must be provided.
Response: Thank you for the comment. The pore size distribution has been included in the inset of Figure 2b.
- Page 8, line 306-310, lines are highlighted.
Response: Thank you for the comment. The highlighted lines have been removed.
- Line 217-219- the sentence need to be revised.
Response: Thank you for the comment. The sentence has been revised. The new sentence is highighted in yellow.
- Tables and equation in the manuscript must be presented with respect to journal format.
Response: Thank you for the comment. The tables and equations have been revised according to the recommended format.
- Regarding the low visible light irradiation, authors have mentioned 48 watts in abstract and in page 3 authors have mentioned 43.8 W/m-2. Authors have to describe the light intensity in mW/cm2.
Response: Thank you for the comment. The light intensity has been corrected accordingly to mW/cm2 throughout the manuscript.
- Authors have to clearly differentiate between LTO/TiO2 and LTO composite in the manuscript.
Response: Thank you for the comment. Section 3.1.1 has been rewritten to give a better understanding of LTO/TiO2 and LTO.
- Electrochemical impedance spectra measurements must be carried out to prove the conductivity of the materials.
Response: Thank you for the comment. We currently do not have such testing facilities and our focus is not in electrochemical measurement.
- XPS spectra of bare ZnO, LTO/TiO2 spectra must be compared. More detailed spectra of composite must be provided with appropriate legend and discussed clearly.
Response: Thank you for the comment. Since the main focus of this research is on the ZnO/LTO/TiO2, the XPS analysis of bare ZnO and LTO/TiO2 were not done. The XPS analysis of ZnO/LTO/TiO2 is able to provide comprehensive information on the chemical state of Zn, O, Li and Ti required to achieve the objective of this project.
However, we have included the XRD of bare ZnO (refer Figure 1a) and confirmed its crystalline structure to be hexagonal wurtzite structure (ICDD file #98-002-9272). Section 3.11 has been rewritten to include the XRD analysis of ZnO. Section 3.2.4 has been rewritten to provide a better understanding of the XPS results.
- During the results presentation, please provide a comparison with previous similar materials. Why this catalyst should be better than the previous catalysts?
Response: Thank you for the comment. The comparisons have been made and discussed in page 13, last paragraph. A few examples of catalysts reported in the literature are shown in Table 3. The changes are highlighted in yellow.
- Figure 4 x-axis unit must be revised.
Response: Thank you for the comment. The x-axis of Figure 4 has been revised.
- The English of the manuscript must be improved before resubmission. The author should pay particular attention to English grammar, spelling, and sentence structure so that the expression of the study is clear to reader.
Response: Thank you for the comment. The manuscript has been revised for language accordingly as recommended.
- Explain better the mechanism of MB degradation. Why by incorporating the ZnO to LTO/TiO2, the degradation was improved? What about the mechanism of degradation when ZnO and LTO were used alone?
Response: Thank you for the comment. The contribution of ZnO in enhancing the photocatalytic activity is discussed in Section 3.2, page 9, line 388-399. The mechanism of degradation for ZnO and LTO was not discussed in details since it was not the main objective of this research.
- More related papers should be cited/referrred 3390/coatings10050500; 10.1016/j.jphotochem.2019.02.020; 10.1016/j.jallcom.2020.154281; 10.1016/j.colsurfa.2019.124294; 10.1088/1361-6528/ab268a/meta; 10.1016/j.optmat.2018.05.022
Response: Thank you for the suggestion. Additional papers and some other new papers have been included in the manuscript, as suggested. The new references are highlighted as well.
The authors wish to acknowledge Reviewer #2 for the insightful and constructive comments on the above manuscript. We have further edited the manuscript for language, clarity, and syntax throughout to meet the high publication standards of this journal.

Round 2
Reviewer 1 Report
The manuscript can be published now.